



# Wave boundary layer model in SWAN revisited

Jianting Du[1,3,4], Rodolfo. Bolaños[2], Xiaoli Guo Larsén[1], and Mark Kelly[1]

[1]Department of Wind Energy, Technical University of Denmark, Risø Campus, Roskilde, Denmark
[2]DHI, Agern Allé 5, DK-2970 Hørsholm, Denmark
[3]First Institute of Oceanography, State Oceanic Administration, Qingdao, China
[4]Laboratory for Regional Oceanography and Numerical Modeling, Qingdao National Laboratory for Marine Science and Technology, Qingdao, China

**Correspondence:** Jianting Du (dujt@fio.org.cn)

**Abstract.** In this study we extend the work presented in Du et al. (2017) to make the WBLM applicable for real cases by improving the wind input and white-capping dissipation source functions. Improvement via the new source terms includes three aspects. First, the WBLM wind-input source function is developed by considering the impact of wave-induced wind profile variation on the estimation of wave growth rate. Second, the white-capping dissipation source function is revised to

5  be not explicitly dependent on wind speed for real wave simulations. Third, several improvements are made to the numerical WBLM algorithm, which increase the model's numerical stability and computational efficiency. The improved WBLM wind-input and white-capping dissipation source functions are calibrated through idealized fetch-limited and depth-limited studies, and validated in real wave simulations during two North Sea storms. The new WBLM source terms show better performance in the simulation of significant wave height and mean wave period than the original source terms.

## 1 Introduction

The accuracy of spectral ocean wave models depends on the forcing from wind, water level, currents, etc. It also depends on the source terms and numerical methods (Ardhuin, 2012). In deep water conditions, the source terms are reduced to wind-input source function ($S_{in}$), wave-breaking dissipation source function ($S_{ds}$), and nonlinear four-wave-interaction source function ($S_{nl}$). In a previous study (Du et al., 2017), a wave boundary layer model (WBLM) was implemented in the third generation

15  ocean wave model SWAN (Booij et al., 1999) to improve the wind-input source function of Janssen (1991, hereafter JANS); this was done by considering the momentum and kinetic energy conservation at each level in the wave boundary layer. It was shown that the new $S_{in}$ improves wave simulations in idealized fetch-limited conditions. Because the evolution of wave spectrum depends on the difference between source and sink terms, the change of $S_{in}$ has to be followed by the tuning of the parameters in $S_{ds}$ (Cavaleri, 2009). Du et al. (2017) simply re-calibrated the white-capping dissipation parameters of Komen

20  et al. (1984, here after KOM) to be proportional to the WBLM $S_{in}$ (Babanin et al., 2010), and wind speed at 10 m ($U_{10}$) (Melville and Matusov, 2002). Such a method works in idealized fetch-limited conditions when the winds do not change over time. However, in real cases, wind speed and direction vary in time. Also, wave breaking is related to wave properties such as wave steepness, rather than explicitly depending on wind speed (e.g. G. J. Komen et al., 1994). Moreover, in coastal areas,





the bottom friction and depth-induced breaking dissipation become important and they influence the shape of wave spectrum. Consequently $S_{in}$ and $S_{ds}$ are also modified by the shallow-water effect. Therefore the description of the new $S_{in}$ and $S_{ds}$ in shallow water also needs to be investigated, before they are used in real simulations.

 Theoretical models of wave-breaking dissipation have been extensively reviewed by G. J. Komen et al. (1994), Young and
5 Babanin (2006a), and Cavaleri et al. (2007) and can be classified into: white-capping models (Hasselmann, 1974), saturation-based models (e.g. Phillips, 1985), probabilistic models (e.g. Longuet-Higgins, 1969; Yuan et al., 1986; Hua and Yuan, 1992), and turbulent models (Polnikov, 1993). Among them, white-capping and saturation-based models are widely used in ocean wave models such as WAM (Komen et al., 1994), SWAN (Booij et al., 1999), WAVEWATCH III (Tolman and Chalikov, 1996), and MIKE 21 SW (Sørensen et al., 2004). White-capping models consider the effect of downward-moving whitecaps
10 doing work against the upward-moving waves. Parameterization of white-capping dissipation can be found in e.g Komen et al. (1984), Bidlot et al. (2007), and Bidlot (2012); the dissipation at all frequencies is taken to be proportional to the mean wave steepness defined by a mean wave number and the significant wave height. The saturation-based models assume saturation exists in the equilibrium range of the wave spectrum, and the dissipation rate is proportional to the saturation at any given frequency. Therefore, the dissipation at each frequency is proportional to the local wave steepness or local saturation.
15 Latter studies, however, suggest a two-phase behavior of wave-breaking dissipation (Donelan, 2001; Babanin and Young, 2005; Young and Babanin, 2006a): $S_{ds}$ should be a function of the spectral peak plus a cumulative frequency-integrated term at higher frequencies due to dominant wave-breaking. Considering the complexity of wave-breaking processes, recent studies tend to combine the two types of $S_{ds}$ together. Alves and Banner (2003) and van der Westhuysen et al. (2007) used a saturation-based model multiplied by a KOM-shaped model, to account for the long-wave-short-wave and wave-turbulence interactions. Banner
20 (2010) introduced a breaking probability function to the saturation-based model of Phillips (1985). Ardhuin et al. (2010), Babanin et al. (2010), and Zieger et al. (2015) added a cumulative term to a saturation-based model. Such combined $S_{ds}$ are proved to be robust in wave simulations, globally to coastal areas (Ardhuin et al., 2012; Ardhuin and Roland, 2012; Leckler et al., 2013).

 However, as more physical processes are being taken into account, expressions of $S_{ds}$ become more complex and need
25 more tuning parameters; e.g. the $S_{ds}$ of Ardhuin et al. (2010) needs up to 18 parameters, which makes it difficult to adjust when there is modification of other source terms. The present study aims at finding a proper dissipation source function that is suitable for the new WBLM $S_{in}$. Therefore, instead of introducing more physics into $S_{ds}$, numerical adjustment is applied to the KOM dissipation (Komen et al., 1984). The reason that we chose the KOM $S_{ds}$ is that it has been shown successful when used with different wind-input source functions in SWAN (Snyder et al., 1981; Komen et al., 1984; Janssen, 1991; Larsén
30 et al., 2017), and because the formulation is flexible such that its total magnitude and spectral distribution can be easily tuned with the parameters. Du et al. (2017) has shown that numerical adjustment to the KOM $S_{ds}$ can be used for the WBLM $S_{in}$, to reproduce the fetch-limited wave growth curve of Kahma and Calkoen (1992). Moreover, Ardhuin (2012) showed that $S_{ds}$ of the KOM type and saturation-based type (Phillips, 1985) can be adjusted to give very similar behavior. However, we found that using only the KOM $S_{ds}$ within the WBLM produces too high energy level at frequencies higher than the spectral peak
35 ($f > f_p$), and this problem can be solved by using a cumulative dissipation term according to Ardhuin et al. (2010). In this





paper, the improvement of WBLM $S_{in}$, the revised $S_{ds}$, and the numerical algorithm changes to the model are presented in section 2. Then the new pair of $S_{in}$ and $S_{ds}$ is calibrated in idealized fetch-limited and depth-limited study, and validated in real case storm simulations in the North Sea. These numerical experiments are described in section 3. And the results are presented in section 4. Wave parameters such as significant wave height, mean period, peak wave period, and spectral shape

are validated using point measurements of deep and shallow waters. $S_{in}$ and $S_{ds}$ of KOM and JANS are also examined as benchmark reference for these storms.

## 2  Methods

### 2.1  The wind-input source function

According to Du et al. (2017), the growth rate ($\beta_g$) of the WBLM wind-input source function ($S_{in} = \beta_g(\sigma,\theta)N(\sigma,\theta)$) is

expressed as:

$$\beta_g(\sigma,\theta) = C_\beta \sigma \frac{\tau_t(z)}{\rho_w c^2} cos^2(\theta - \theta_w), \tag{1}$$

where $N(\sigma,\theta)$ is the action density spectrum, $\theta$ and $\theta_w$ is the wave and wind direction, $C_\beta$ is the Miles' parameter (Miles, 1957), $\rho_w$ is the water density, and $c$ is the phase velocity of waves. $\tau_t(z)$ is the local turbulent stress, which equals to the total stress, $\tau_{tot}$, minus the wave-induced stress, $\tau_w(z)$. The Miles' parameter $C_\beta$ is described as a function of the non-dimensional critical height, $\lambda$:

$$C_\beta = \frac{J}{\kappa^2} \lambda \ln^4 \lambda, \lambda \leq 1, \tag{2}$$

where $\kappa = 0.41$ is the von Kármán constant, and $J = 1.6$ is a constant. In Du et al. (2017), the expression of the non-dimensional critical height $\lambda$ for Miles' parameter (equation 2) is derived by the assumption of a logarithmic wind profile followed Janssen (1991), and it is expressed as:

$$\lambda = \frac{gz_0}{c^2} \exp\left[\frac{\kappa}{(u_*/c + \alpha) \cdot \cos(\theta - \theta_w)}\right], \tag{3}$$

where $g$ is the gravity acceleration, $\alpha = 0.008$ is a wave age tuning parameter according to Bidlot (2012), $z_0$ is the roughness length. However, it is found that using equation (3) causes numerically instability in some cases. This is because within the

WBL, the wind profile is not logarithmic under the impact of waves (Du et al., 2017). Using a logarithm wind profile (equation 3) not only slows down the computation but could also fails in converging in some cases. Therefore when applying WBLM $S_{in}$, the expression of $\lambda$ also needs to be changed to adjust to the new wind profile. Here we follow Miles (1957)'s procedure to drive an approximate expression for $\lambda$. In Miles (1957) the non-dimensional critical height is defined as:

$$\lambda = kz_c, \tag{4}$$

where $k$ is the wave number, $z_c$ is the critical height where the wave phase velocity ($c$) equals the wind speed, $u(z_c)$. Consid-

ering the misalignment of wind and wave direction, we have

$$c = u(z_c) \cdot \cos(\theta - \theta_w). \tag{5}$$



We assume that in the vicinity of the critical height ($z_c$), the wind profile can be approximately described as locally logarithmic

$$\frac{du}{dz} = \frac{u_*^l}{\kappa z}, \tag{6}$$

where $u_*^l = \sqrt{\tau_t/\rho_a}$ is the local friction velocity. In the vicinity of the critical height, wind speed at any other heights $z$ can be expressed as

$$u(z) = \frac{u_*^l}{\kappa} \ln(z) + z_0^l, \tag{7}$$

where $z_0^l$ is a local effective roughness. Introducing equation (7) to equation (5), we have wind speed at the critical height

$$u(z_c) = \frac{c}{\cos(\theta - \theta_w)} = \frac{u_*^l}{\kappa} \ln(z_c) + z_0^l. \tag{8}$$

The critical height is calculated by combining equation (7) and (8)

$$z_c = z \cdot \exp\left[\frac{\kappa}{(u_*^l/c) \cdot \cos(\theta - \theta_w)} - \frac{\kappa u(z)}{u_*^l}\right]. \tag{9}$$

Considering the shallow water dispersion relation, $k = (g/c^2)\tanh(kh)$ with $h$ the water depth, the combination of equation (4) and (9) results in the non-dimensional critical height for any direction

$$\lambda = k z_c = \frac{gz}{c^2}\tanh(kh) \cdot \exp\left[\frac{\kappa}{(u_*^l/c) \cdot \cos(\theta - \theta_w)} - \frac{\kappa u(z)}{u_*^l}\right]. \tag{10}$$

It is found that equation (10) tends to underestimate wave growth at low frequencies. Following WAM (https://github.com/mywave/WAM), a wave age tuning parameter $\alpha = 0.011$ is added to increase wave growth at low frequencies:

$$\lambda = k z_c = \frac{gz}{c^2}\tanh(kh) \cdot \exp\left[\frac{\kappa}{(u_*^l/c + \alpha) \cdot \cos(\theta - \theta_w)} - \frac{\kappa u(z)}{u_*^l}\right]. \tag{11}$$

## 2.2  White-capping dissipation source function

The white-capping dissipation expression of KOM (Komen et al., 1984; Janssen, 1991; Bidlot et al., 2007) in SWAN is written as:

$$S_{ds}(\sigma,\theta) = -C_{ds}\langle\sigma\rangle\left(\langle k\rangle^2 m_0\right)^2\left[(1-\Delta)\frac{k}{\langle k\rangle} + \Delta\left(\frac{k}{\langle k\rangle}\right)^2\right]\phi(\sigma,\theta), \tag{12}$$

where $\phi(\sigma,\theta) = \sigma N(\sigma,\theta)$ is the frequency spectra. In this study, the mean radian frequency $\langle\sigma\rangle$ and mean wave number $\langle k\rangle$ is modified according to Bidlot et al. (2007) to put more emphasis on the high frequencies

$$\begin{cases} \langle\sigma\rangle = \iint \sigma\phi(\sigma,\theta)\,d\theta d\sigma/m_0 \\ \langle k\rangle = \left[\iint k^{1/2}\phi(\sigma,\theta)\,d\theta d\sigma/m_0\right]^2, \end{cases} \tag{13}$$





where $m_0 = \int \int \phi(\sigma, \theta)\, d\theta d\sigma$ is the variance of the sea surface elevation. The choice of the two dissipation parameters, $C_{ds}$ and $\Delta$, are different for different wind-input source functions. For example, for KOM $S_{in}$, $C_{ds} = 2.5876$, $\Delta = 1$; for JANS $S_{in}$, $C_{ds} = 4.5$, $\Delta = 0.5$; for WBLM $S_{in}$ in Du et al. (2017), $\Delta = 0.1$, and $C_{ds}$ in $S_{ds}$ is related to $S_{in}$ to make sure

$$\int S_{ds}(\sigma)\, d\sigma = R_{ds} \int S_{in}(\sigma)\, d\sigma, \tag{14}$$

where

$$R_{ds} = 1 - 0.15 \left( \frac{10}{U_{10}} \right)^{0.5} \cdot \max \left[ 1.0, 1.53 \left( \frac{5.2 \times 10^{-7}}{\widetilde{E}} \right)^{0.25} \right], \tag{15}$$

where $U_{10}$ is wind speed at 10 m, $\widetilde{E} = m_0 g^2 / U_{10}^4$ is a non-dimensional energy. As discussed in the introduction, a dissipation parameter that is strongly dependent on wind speed as in equation (15) only works in idealized fetch-limited cases but will in principle not work in real cases because wave breaking depends on wave state rather than wind. Here we explore the use of some wave parameters to replace $U_{10}$ and $S_{in}$ in equations (14) and (15) to get rid of the direct dependence on wind speed. We derive the relationship between $U_{10}$, $m_0$, peak frequency ($f_p$), and fetch ($x$) from the three non-dimensional parameters, namely non-dimensional energy ($\widetilde{E}$), non-dimensional peak frequency ($\widetilde{F}_p = f_p U_{10}/g$), and non-dimensional fetch ($\widetilde{x} = xg/U_{10}^2$). The fetch dependence of $\widetilde{E}$ and $\widetilde{F}_p$ can be written as:

$$\begin{cases} \widetilde{E} = A\widetilde{x}^B \\ \widetilde{F}_p = C\widetilde{x}^D, \end{cases} \tag{16}$$

where in Kahma and Calkoen (1992) (composite condition), $A = 5.2 \times 10^{-7}$, $B = 0.9$, $C = 2.1804$, $D = -0.27$. According to equation 16, $U_{10}$, $\widetilde{E}$, and $\widetilde{x}$ can be expressed as functions of $m_0$, $f_p$, and $g$:

$$\begin{cases} U' = \left[ \frac{C^B}{A^D} \frac{m_0^D}{f_p^B} g^{2D+B} \right]^{\frac{1}{4D+B}} \\ E' = m_0 g^2 / U'^4 \\ x' = (E'/A)^{1/B}, \end{cases} \tag{17}$$

where $U_{10}$, $\widetilde{E}$, and $\widetilde{x}$ are replaced by $U'$, $E'$, and $x'$ since they are parameterized variables. The dissipation coefficient $C_{ds}$ in equation (12) can be obtained by fitting the $C_{ds}$ calculated from equation (14) and (15) with $U'$ and $x'$ from equation (17):

$$C_{ds} = F(x', U'). \tag{18}$$

The form and parameters in equation (18) will be obtained in the fetch-limited study in section 4.1. To increase the robustness of the wave modeling in case of unusual shaped spectra, the peak frequency $f_p$ in equation (17) is replaced by $0.866\langle f \rangle$ according to Komen et al. (1994) who uses $k_p = 0.75\langle k \rangle$, where $\langle f \rangle = \langle \sigma \rangle / 2\pi$ is the mean frequency.



To reduce the energy level at high frequencies, a cumulative term is added to the dissipation source functions. The cumulative dissipation term ($S_{ds}^c$) follows Ardhuin et al. (2010), but the directional dependence of dissipation rate is not considered:

$$S_{ds}^c (f,\theta) = -1.44 \times C_{cu}\phi(f,\theta) \int\limits_{0}^{r_{cu}f} \max\left[\left(\sqrt{B(f')} - \sqrt{B_r}\right),0\right]^2 |c-c'|' \, df',$$ (19)

where $C_{cu} = 1.0$ is a dissipation parameter, $B_r = 0.0012$ is a saturation threshold, $r_{cu} = 0.5$ is the ratio of the maximum frequency where dissipation of long waves influence short waves, $C_g$ is the group velocity, $B(f)$ is the local saturation (van der Westhuysen et al., 2007):

$$B(f) = \int\limits_{0}^{2\pi} k^3 \cos^2(f,\theta') \phi(f,\theta') \frac{C_g}{2\pi} d\theta'.$$ (20)

## 2.3 Improvement on the numerical algorithm

Considering the expensive cost of WBLM code in Du et al. (2017), improvement on the numerical algorithm of the WBLM (Du et al., 2017) was done in the following aspects.

- Reducing the unnecessary calculations in the high frequencies. The WBLM uses 10 Hz as the maximum frequency, which is only being used for very young waves. Usually, the WBLM does not have to solve such high frequencies when there is no energy contained in that range. Therefore, in the new code, the WBLM only solve the energy containing frequency range which is dynamically changing with the wave spectrum. Such an adjustment reduces approximately half of the computation time in the idealized fetch-limited study in section 3.1.

- The standard calculation in SWAN, a sweeping technique is used for the directional propagation of the waves, which needs four times sweep for each time step. Such sweep is not necessary for the calculation of WBLM because the WBLM has to integrate over all directions of the spectrum. Therefore, WBLM only calculates once per time step.

With the above mentioned refinement, the WBLM is now about 5 times faster than the previous version in Du et al. (2017), and uses similar computation time as the KOM and JANS formulations.

## 3 Experiments

### 3.1 Idealized fetch-limited study

The revised dissipation parameter (equation 18) in $S_{ds}$ together with the new non-dimensional critical height as in WBLM $S_{in}$ were first calibrated in the idealized fetch-limited wave growth experiments with the same model setup as in Du et al. (2017). Here we briefly describe the model setup. We use the one-dimensional SWAN. The fetch is between 0 and 3000 km, with the spatial resolution changes gradually from 0.1 km to 10 km. The wave spectrum ranges from 0.01 to 10.51 Hz, and the frequency



discretization was logarithmic with a frequency exponent of 1.1, which result in 73 frequencies. We use 36 directional bins with a constant spacing of $10°$. SWAN initializes from zero spectrum and runs for 72 h with a time step of 1 min. $U_{10}$ ranges from 5 to 40 ms$^{-1}$ and keeps constant during the 72 hours simulation.

5     The calibration process is carried out in three steps. Step 1, we run the model using the white-capping dissipation parameter as in Du et al. (2017) (equation 14 and 15) in the idealized fetch-limited study. Since we added a cumulative dissipation source term, the parameters in equation 15 has to be changed into equation 21 so as to best fit to the Kahma and Calkoen (1992) fetch-limited wave growth curves

$$R_{ds} = 1 - 0.18 \left( \frac{10}{U_{10}} \right)^{0.3} \cdot \max \left[ 1.0, 1.53 \left( \frac{5.2 \times 10^{-7}}{\widetilde{E}} \right)^{0.25} \right]. \tag{21}$$

In Step 2, the real form and parameters in equation 18 will be obtained by analyzing and fitting the $C_{ds}$ with $x'$ and $U'$, which are calculated in Step 1. Finally, the best fit in Step 2 may not be the best choice for the wave model. Therefore, the final choice 10 of the $C_{ds}$ equation is decided through idealized fetch-limited study with several chosen fitting parameters.

## 3.2   Idealized depth-limited study

In addition to the idealized fetch-limited study, the revised WBLM source terms are also tested in depth-limited wave growth experiments, to check if the new source terms perform well with the interaction of the other source terms in the wave model, including the bottom friction and depth-induced wave breaking dissipation source functions. Following SWAN (2014), we use 15 JONSWAP (Hasselmann et al., 1973) bottom friction with a bottom friction coefficient, $C_b = 0.038$ m$^2$s$^{-3}$, and Battjes and Janssen (1978) depth-induced wave breaking source function with a breaker parameter, $\gamma = 0.8$.

    In the depth-limited wave growth experiments, we take the measurements of Young and Babanin (2006b) as reference for validation, because they not only provided detailed wind, wave, and water depth information, but also provided wave spectrum measurement from capacitance wave probes up to 10 Hz (Young et al., 2005). Zijlema et al. (2012) did similar depth-limited 20 wave growth experiments for the calibration of the bottom friction parameter in SWAN, but they did not compare the wave spectrum. Zijlema et al. (2012) selected 10 representative cases from the data presented in Young and Babanin (2006b). We add 3 more cases because the wave spectrum in these three cases are also presented in Young and Babanin (2006b). This ends up with 13 cases in all, which are number 1, 11, 17, 23, 28, 30, 32, 58, 61, 63, 82, 83, and 87 in Table 1 of Young and Babanin (2006b). Among them, number 1, 11, 28, 32, and 87 have wave spectrum records for model validation. In all the 13 cases, the 25 water depth ranges from 0.89 m to 1.1 m, and the wind speed ranges from 5.7 ms$^{-1}$ to 15 ms$^{-1}$. The fetch is set to 20 km with a spatial resolution of 0.1 km to ensure the fetch is long enough, and the wave growth is limited by the water depth. The frequency and directional discretization are same as the fetch-limited study. SWAN initializes from zero spectrum and runs for 24 h (we found 24 h is long enough for the wave development in the shallow water conditions in this study), with a time step of 1 min. Two pair of $S_{in}$ and $S_{ds}$ are tested, namely KOM (Snyder et al., 1981; Komen et al., 1984) and the revised WBLM 30 in this study.





### 3.3 Real case study in the North Sea

The new WBLM $S_{in}$ and $S_{ds}$ are validated during two winter storms in the North Sea. Wind and wave measurements are obtained during the "Reducing the uncertainty of near-shore wind estimations using wind lidars and mesoscale models" (RUNE) Project (Floors et al., 2016b). Simultaneous wind and wave measurements from lidar and buoy are available from Nov. 2015
to Jan. 2016. The experiment was conducted at the west coast of Jutland, Denmark, with an average water depth of 16.5 m. Details about the wind and wave measurements can be found in (Floors et al., 2016b, a, c; Bolaños, 2016; Bolaños and Rørbæk, 2016). Beside the standard wave parameters such as significant wave heights, peak wave period, and mean wave periods, the two-dimensional wave spectrum $E(f,\theta)$ are also available, which allows us to validate more detailed aspects of the source functions. During the RUNE period, two storms passed the measurement site from 2015-11-28 to 2015-12-08. Wave
simulations were done during this period with the three pair of source terms (KOM, JANS, and WBLM). Beside the measurement from the RUNE project, point wave measurements at Fjaltring, Hanstholm, A121, Vaderoarna, and Helgoland North from NOOS (https://noos.bsh.de/); FINO-1 and FINO-3 from FINO (http://fino.bsh.de) at; and Sleipner-A and Ekofisk from eKlima (http://sharki.oslo.dnmi.no) during the simulation period are also used for model validation. The locations of these measurement sites are shown in Figure 1a, b and c.
SWAN is forced by the NCEP Climate Forecast System version 2 (CFSR) 0.312-degree resolution 10 m wind. SWAN uses three nested domain, with a resolution downscaling from 9 km to 3 km and 600 m (Figure 1). The bottom friction and depth-induced wave breaking source functions for the real case study are same as the depth-limited wave growth studies. Bathymetry data is obtained from the EMODnet Digital Terrain Model (DTM) with a spatial resolution of 1/8 arc-minute. Open boundaries for SWAN are set to zero. The frequency discretization was logarithmic with a frequency exponent of 1.1, and the lowest
frequency was set to 0.03 Hz. For KOM and WBLM source terms, a cut-off frequency of 10.05 Hz is used, giving a total number of 61 frequencies; for JANS source terms, the cut-off frequency is set to 0.57 Hz to make sure the simulation stable, giving a total number of 31 frequencies. We used 36 directional bins and 5 min time step. SWAN initializes from zero spectrum and the first 24 hours output are not included in our analysis.

A summary of the constants and model setups used for all the experiments in section 3 are listed in Table 1 and Table 2,
respectively.

**Table 1.** Constants used for all the experiments in section 3. $C_{ds}$ and $\Delta$ are white-capping dissipation parameters in equation (12); $F(x',U')$ is the new dissipation parameter for WBLM in equation (18); $C_{cu}$, $B_r$, and $r_{cu}$ are cumulative dissipation parameter for WBLM in equation (19); $C_b$ and $\gamma$ are JONSWAP (Hasselmann et al., 1973) bottom friction and Battjes and Janssen (1978) depth-induced wave breaking parameters.

|       | $C_{ds}$    | $\Delta$ | $C_{cu}$ | $B_r$  | $r_{cu}$ | $C_b\left(m^2 S^{-3}\right)$ | $\gamma$ |
|-------|-------------|----------|----------|--------|----------|------------------------------|----------|
| KOM   | 2.5876      | 1.0      | –        | –      | –        | 0.038                        | 0.8      |
| JANS  | 4.5         | 0.5      | –        | –      | –        | 0.038                        | 0.8      |
| WBLM  | $F(x',U')$  | 0.1      | 1.0      | 0.0012 | 0.5      | 0.038                        | 0.8      |



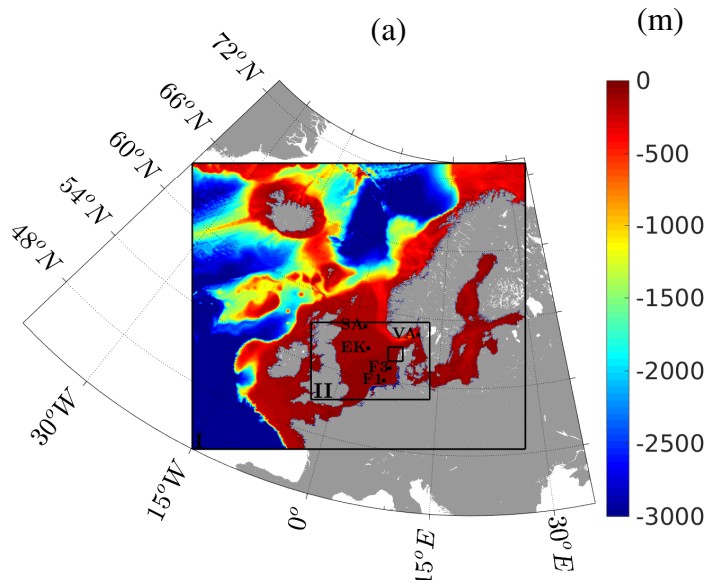

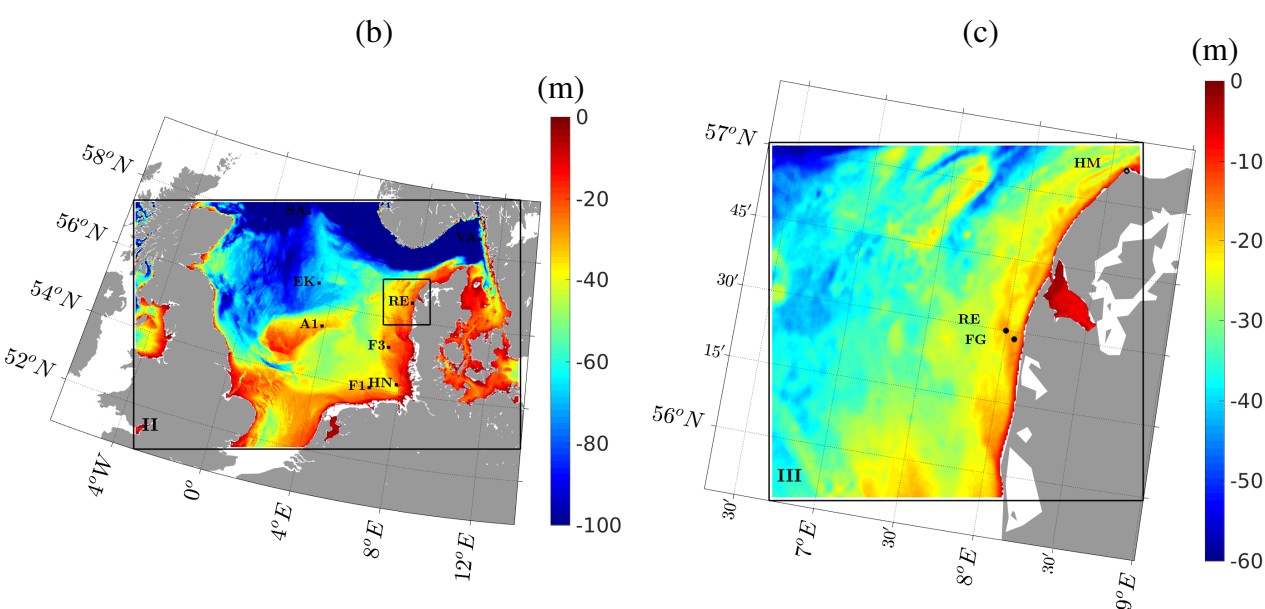

**Figure 1.** SWAN domain for RUNE storm simulation, with domain I 9 km resolution, II 3 km resolution, and III 600 m resolution. (a), (b), and (c) show the bathymetry at domain I, II, and III. Bathymetry are interpolated from EMODnet Digital Terrain Model (DTM) 1/8 arc-minute data. The 5 measurement sites, RUNE (RE), Fjaltring (FG), Hanstholm (HM), A121 (A1), Vaderoarna (VA), Helgoland North (HN), Fino-1 (F1), Fino-3 (F3), Sleipner-A (SA), and Ekofisk (EK), are shown as black dots.


**Table 2.** A summary of model setups for all the experiments in section 3. $dx$ is the spatial resolution and $dt$ is the time step of SWAN in seconds.

| Experiments | Simulation time | $U_{10}$(ms$^{-1}$) | Bathymetry | $dx$ (km) | $dt$ (sec) | Frequency (Hz) | Direction bins |
|---|---|---|---|---|---|---|---|
| Fetch-Limitted | 72 h | 5.0 - 40 | 5000 m | 0.1 - 10 | 60 | 0.001 - 10.51 | 36 |
| Depth-Limitted | 24 h | 5.7 - 15 | 0.89 - 1.1 m | 0.1 | 60 | 0.001 - 10.51 | 36 |
| RUNE storms | 2015-11-28 to 2015-12-08 | CFSR | EMODnet DTM | 9-3-0.6 | 300 | 0.003 - 10.05 (0.57 for JANS) | 36 |

## 4 Results

### 4.1 Idealized fetch-limited study

The significant wave height ($H_{m0}$) calculated from Step 1 of the idealized fetch-limited study using equation 14 and 21 are shown in Figure 2a. The Kahma and Calkoen (1992) wave growth curves are also shown as solid black lines for reference. Note

that Kahma and Calkoen (1992) curves come from data with limited wind speeds ($U_{10} \le 25$ ms$^{-1}$) and fetches ($x \le 300$ km), and we linearly extend them to broader ranges. The $H_{m0}$ calculated from Step 1 (solid colored lines in Figure 2a) agree with Kahma and Calkoen (1992) wave growth curves for fetches $x \le 10$ km. But it tends to underestimate $H_{m0}$ for fetches $x > 10$ km. Therefore, in Step 2 we only fit the $C_{ds}$ in the first 10 km and extend its application for longer fetches. The $C_{ds}$ calculated from Step 1 for wind speed $U_{10} = 5$ ms$^{-1}$ and $U_{10} = 20$ ms$^{-1}$ are shown in Figure 2b as black circles and black rectangles.

Here we try to surmise the form of equation 18 based on the distribution of $C_{ds}$ from Step 1 in Figure 2b. First, $C_{ds}$ has a clear dependence on $U_{10}$. Taking 10 ms$^{-1}$ as reference, there should be a $\left( \frac{U'}{10} \right)$ term in the $C_{ds}$ equation. Second, $C_{ds}$ depends on the fetch, considering the fetch dependence is logarithmic (the horizontal coordinate of Figure 2b is logarithmic), there should be a $ln(x)$ term in the $C_{ds}$ equation. Considering that a log transformed quantity must be unitless, we use the non-dimensional fetch $ln(x')$ instead of $ln(x)$. Therefore, we assume $C_{ds}$ in equation 18 has the following form:

$$C_{ds} = [C1 \cdot \ln^{C2}(x') + C3] \cdot \left( \frac{U'}{10} \right)^{C4}, \tag{22}$$

where $C1$, $C2$, $C3$, and $C4$ are parameters to be determined by fitting the $C_{ds}$ calculated from Step 1. To reduce the number of the fitting parameters, we use fixed value for the power on $ln(x')$ and $\left( \frac{U'}{10} \right)$. By testing $1 \le C2 \le 4$ with a resolution of 1, and $0 \le C4 \le 1$ with a resolution of 0.1, we choose $C2 = 2$ and 4, $C4 = 0.5$, and 1 as representative values. With the values of $C2$ and $C4$ provided, we fit $C1$ and $C3$ with the data from Step 1 and end up with the first 3 groups of fitting parameters listed in Table 3 (FIT1 to FIT3). FIT4 in Table 3 is an improvement for FIT3 which will be described latter. The fitted $C_{ds}$ using FIT1 to

FIT4 at different wind speed conditions are shown in Figure 2b as colored lines with circles and rectangles representing wind speed.

  The 4 groups of parameters (FIT1 to FIT4 in Table 3) are applied in SWAN in the fetch-limited study, and the results are shown in Figure 2c and 2d. The effect of the $\left( \frac{U'}{10} \right)^{C4}$ term in equation 22 can be seen from the comparison between FIT1 and FIT2. The fitted $C_{ds}$, simulated $H_{m0}$ and $T_p$ of FIT1 and FIT2 are shown in Figure 2b, 2c, and 2d as blue solid and red solid





**Table 3.** Four groups of fitting parameters (FIT1, FIT2, FIT3, FIT4) for equation 22.

|      | $C1$  | $C2$    | $C3$  | $C4$ |
|------|-------|---------|-------|------|
| FIT1 | 30.74 | 4       | 1.169 | 1.0  |
| FIT2 | 83.61 | 4       | 1.605 | 0.5  |
| FIT3 | 13.08 | 2       | 1.241 | 0.5  |
| FIT4 | 23.02 | 1.41255 | 0.0   | 0.0  |

lines with circles and rectangles representing wind speed. Although significant difference of $C_{ds}$ between $C4 = 1.0$ (FIT1) and $C4 = 0.5$ (FIT2) is seen in Figure 2b, the resulted $H_{m0}$ and $f_p$ show relatively small difference. In low wind speed conditions ($U_{10} = 05$ and $10$ ms$^{-1}$), $C4 = 1.0$ (blue solid lines in Figure 2c and 2d) results in larger $H_{m0}$ (smaller $f_p$) than $C4 = 0.5$ (red solid lines in Figure 2c and 2d). In high wind speed conditions ($U_{10} = 20$ and $40$ ms$^{-1}$), $C4 = 1.0$ underestimates $H_{m0}$

(overestimates $f_p$) more than $C4 = 0.5$ in long fetches ($x > 10$ km).

The effect of the $ln(x')^{C2}$ term can be seen from the comparison between FIT2 and FIT3, the red solid and magenta dashed lines in Figure 2b, 2c, and 2d. The impact of $C2$ to $H_{m0}$ and $f_p$ is much weaker than $C4$. Using $C2 = 2$ (magenta dashed lines in Figure 2c and 2d) results in slightly larger $H_{m0}$ (smaller $f_p$) than $C2 = 4$ (red solid lines in Figure 2c and 2d) in long fetches ($x > 10$ km), which results in larger white-capping dissipation, smaller $H_{m0}$, and larger $f_p$.

FIT1-FIT3 tend to underestimate $H_{m0}$ (overestimates $f_p$) in long fetches ($x > 10$ km). In the real case study in section 4.3, it will be shown that such an underestimation of $H_{m0}$ failed in capturing large waves in real storm simulations. Therefore, we continuously reduce the value of $C2$ and $C4$ until the large waves are captured in both idealize fetch-limited study and real case study, which results in the parameters of FIT4 (hereafter WBLM).

Figure 3a shows the wave spectrum from WBLM (solid lines) in $10$ ms$^{-1}$ wind speed condition after 72 h simulation in

comparison with the spectrum parameterization of D85 (Donelan et al., 1985) (dashed lines) and JONSWAP (Hasselmann et al., 1973) (doted lines) with the fetch dependence from Kahma and Calkoen (1992). Detailed equations for D85 and JONSWAP are listed in Appendix A. Our model generally follows the shape of the D85 and JONSWAP spectrum, but it tends to underestimate the energy at low frequencies at fetches $x \geq 10$ km. The D85 spectra at short fetches (e.g. 1 km) has less energy in high frequencies (e.g. $f > 1$ Hz) than in long fetches (e.g. 3000 km). On the contrary, JONSWAP and WBLM spectrum have more

energy in short fetches than long fetches in high frequencies. The D85 spectra has a $f^{-4}$ shape at high frequencies, while JONSWAP has a $f^{-5}$ shape. The high frequency part of WBLM spectra is between $f^{-4}$ and $f^{-5}$. Figure 3b shows the source term distribution of wind-input ($S_{in}$, solid lines), white-capping dissipation ($S_{ds}$, dashed lines), and 5 times cumulative dissipation ($5S_{ds}^c$, doted lines) source functions at different fetches indicated by color legends in Figure 3a. $5S_{ds}^c$ instead of $S_{ds}^c$ is used to better visualize the cumulative dissipation source term. As the waves grow older, the $S_{in}$ in high frequencies

become smaller, and $S_{ds}^c$ become larger, which may explain why the WBLM spectra has more energy in short fetches than in long fetches in high frequencies. Figure 3c shows the corresponding stress distribution within the wave boundary layer. Here we also use $5\tau_w^l$ (doted lines) instead of $\tau_w^l$ for the purpose of visualizing the local wave-induced stress. Short fetch waves contribute more surface stress than long fetch waves in high frequencies, while they contribute less stress than long fetch



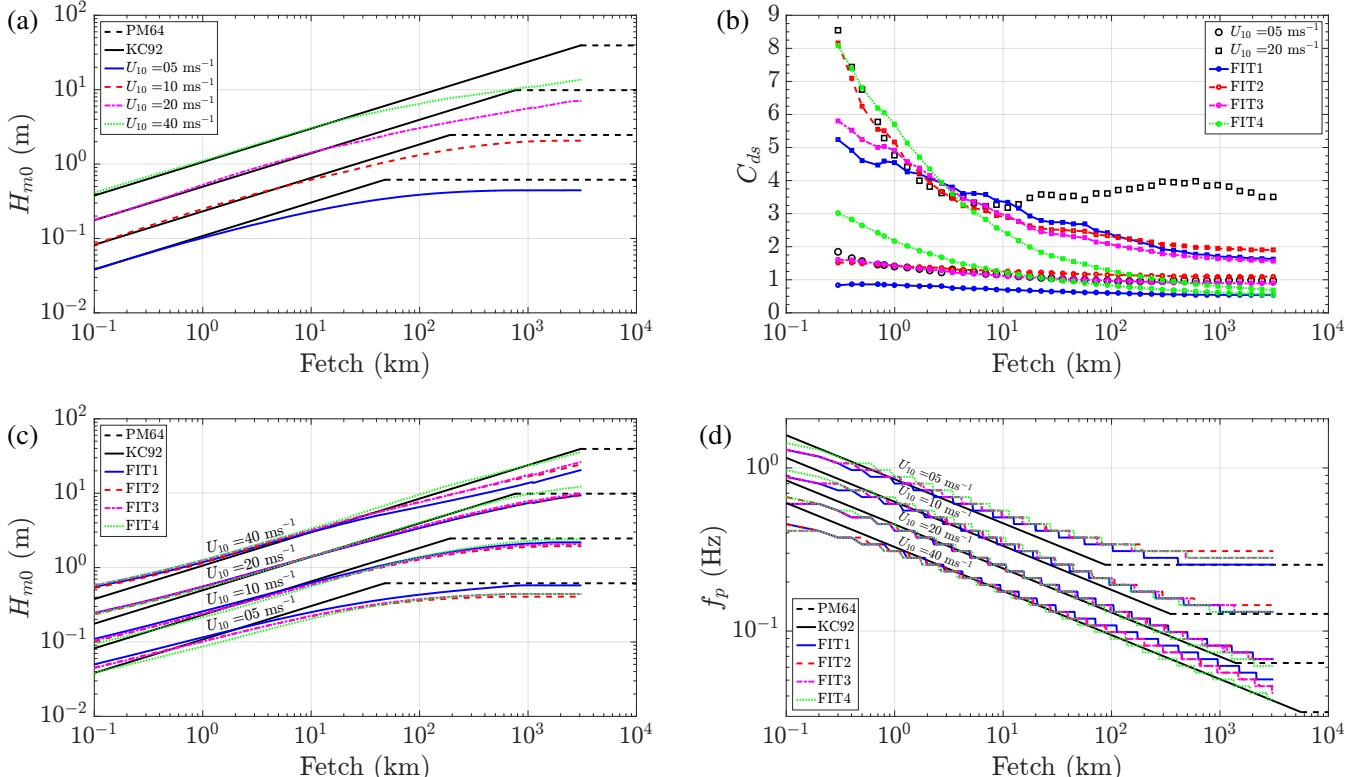

**Figure 2.** $H_{m0}$ as a function of fetch after 72 hours of simulation in different wind speed condition using equation 21 as the white-capping dissipation parameter (panel a). $C_{ds}$ as a function of fetch after 72 hours of simulation in different wind speed condition using equation 21, and fitted $C_{ds}$ using FIT1 to FIT4 (panel b). $H_{m0}$ and $T_p$ as a function of fetch using white-capping dissipation parameters from FIT1 to FIT4 (panel c and d).

waves in low frequencies. The total wave-induced stress depend on the integration of $\tau_w^l$ at all frequencies, which results in waves with fetch of 5 - 15 km have the highest total wave-induced stress, waves with fetch of 1 km have lower wave-induced stress, and waves with fetch of 3000 km have the lowest wave-induced stress. Accordingly, total wind stress ($\tau_{tot}$, thick solid lines) at 5 - 15 km is larger than 1 km and 3000 km because of the impact of the waves. Figure 3d shows the wind

5   profiles within the wave boundary layer calculate by WBLM. Wind profiles are rather different in different wave development stage, which reveals that it is necessary to take the wave-induced wind profile variation into account in the estimation of the critical height in section 2.1.

### 4.2 Idealized depth-limited study

Figure 4 shows the non-dimensional wave energy as a function of non-dimensional depth for fully developed waves in shallow

10   waters after 24 h simulation, with the measurements of Young and Babanin (2006b) as reference. In comparison, results from





**Figure 3.** Wave spectrum (a), wind-input and dissipation source terms (b), stress distribution over frequencies (c), and wind profile (d) calculated from WBLM, in 10 ms$^{-1}$ wind speed condition at different fetches after 72 h simulation, in the fetch-limited wave growth study. The dashed lines and doted lines in panel (a) are the wave spectrum parameterization from D85 (Donelan et al., 1985) and JONSWAP (Hasselmann et al., 1973), the solid lines are from WBLM. The solid, dashed, and doted lines in panel (b) are the WBLM wind-input ($S_{in}$), white-capping dissipation ($S_{ds}$), and 5 times cumulative dissipation ($5S_{ds}^c$) source functions, respectively. Thick solid, thin solid, dashed, and doted lines in panel (c) are the total stress ($\tau_{tot}$), turbulent stress ($\tau_t$), cumulative wave-induced stress ($\tau_w^c$), and 5 times local wave-induced stress ($5\tau_w^l$) from WBLM. The solid lines in panel (d) are calculated from WBLM, and the dashed lines are the relative logarithm wind profiles extended from wind speed at 10 m elevation.



KOM source terms are also shown. Both of the WBLM (red crosses) and KOM (blue squares) show close agreement with the measurements (black circles).

The one-dimensional wave spectrum in the depth-limited experiment is further examined after 24 h simulation in Figure 5a-e for different wind speed and depth conditions, with the measurements of Young and Babanin (2006b) as reference. Both models capture the peak of the wave spectrum. However, KOM (blue lines) tends to underestimate the energy level at high frequencies. On the contrary, the energy level of WBLM (red lines) at high frequencies closely follows the measurements.

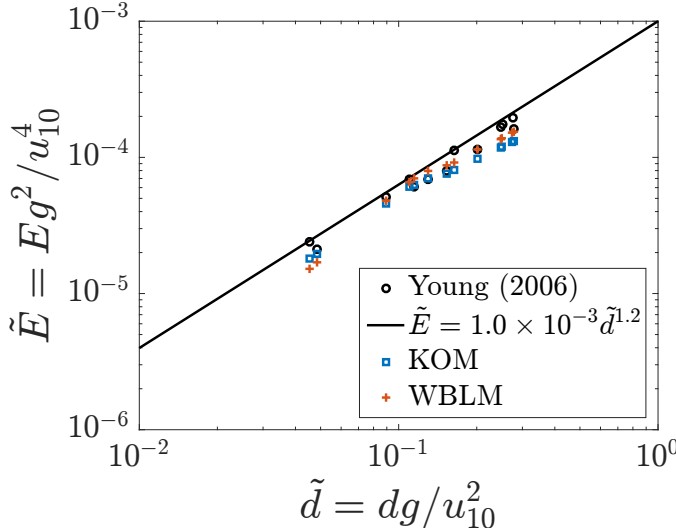

**Figure 4.** Observed (black circles) and parameterized (black line) non-dimensional wave energy for fully developed waves in shallow water as a function of non-dimensional depth (Young and Babanin, 2006b) and SWAN results with KOM (blue squares)and WBLM (red crosses) source terms after 24 h simulation.

### 4.3 Two storms during RUNE project

During the two RUNE storms from 2015-11-28 to 12-08, wave simulation was done with SWAN forced by CFSR wind. The performance of KOM, JANS, and WBLM source terms are evaluated with buoy measurements in terms of significant wave height $H_{m0}$, mean wave direction $D_{mean}$, peak period $T_p$, mean period $T_{m01}$, and one-dimensional wave spectrum. Figure 6 shows the simulated time series of $H_{m0}$, $D_{mean}$, $T_p$, and $T_{m01}$ in comparison with buoy measurements at RUNE. To see the impact of different parameters of the WBLM white-capping dissipation source function to the wave simulation, results from FIT1 to FIT3 are also shown. Similar to the conclusions in the idealized fetch-limited study in section 4.1, these parameters significantly underestimate high waves. Only FIT4 (here WBLM) can be used for real wave simulations to capture the high waves.

Now we compare the performance of the new WBLM with KOM and JANS source terms. For $H_{m0}$, $D_{mean}$, and $T_p$, all the modeled time series generally follow the general trends of measurement data. The biggest error of $H_{m0}$ happens at the




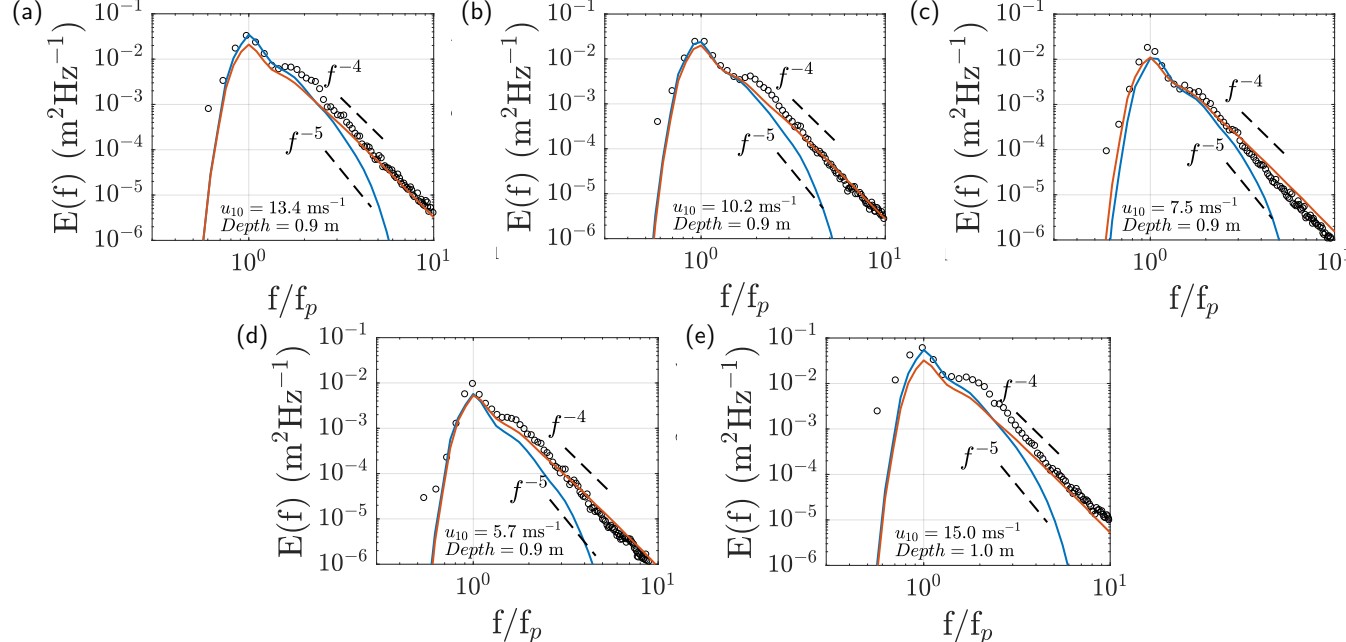

**Figure 5.** One dimensional wave spectrum measured by (Young and Babanin, 2006b) (black circles) for fully developed waves in shallow water and the results from SWAN with KOM (blue lines) and WBLM (red lines) source functions after 24 h simulation.

two storm peaks. The three source terms overestimate the $H_{m0}$ during the peak about 1 m (15%). WBLM gives slightly better $H_{m0}$ during the peak than KOM and JANS. But it tends to underestimate $T_p$ during the storm peaks. WBLM predicts $T_{m01}$ significantly better than KOM and JANS. Note that the Buoy $T_{m01}$ is calculated from the frequency range of 0.005 Hz to 0.64 Hz, JANS is from 0.03 Hz to 0.58 Hz, KOM and WBLM is from 0.03 Hz to 0.63 Hz. A summary of the statistics is listed in

5 Table 4, and the definition of the statistics are given in Appendix B. WBLM generally gives better result for $H_{m0}$ and $T_{m01}$ than KOM and JANS. All the three source terms give similar accuracy in predicting $D_{mean}$. WBLM is slightly less accurate in predicting $T_p$ than KOM and JANS.

**Table 4.** Statistics of simulated significant wave height ($H_{m0}$), mean wave direction ($D_{mean}$), and peak ($T_p$) and mean ($T_{m01}$) wave period in comparison with buoy measurements at RUNE site from 2015-11-28 to 2015-12-08. The statistics include mean difference (BIAS), root mean square difference (RMSE), and scatter index (SI). In each column, the values of smallest absolute errors are signed with bold text.

| Exp. | $H_{m0}(m)$ | | | $D_{mean}$ (°) | | | $T_p(s)$ | | | $T_{m01}$ | | |
|---|---|---|---|---|---|---|---|---|---|---|---|---|
| | BIAS | RMSE | SI | BIAS | RMSE | SI | BIAS | RMSE | SI | BIAS | RMSE | SI |
| KOM | 0.24 | 0.62 | 0.18 | 3.99 | **8.32** | **0.03** | 0.25 | **1.24** | **0.13** | 1.60 | 1.74 | 0.11 |
| JANS | **0.17** | **0.52** | 0.15 | 3.40 | 8.74 | **0.03** | 0.23 | 1.36 | 0.14 | 1.56 | 1.71 | 0.11 |
| WBLM | 0.35 | **0.52** | **0.12** | **2.98** | 8.84 | **0.03** | **-0.13** | 1.44 | 0.15 | **0.57** | **0.67** | **0.06** |



**Figure 6.** Time series during two winter storms in RUNE project. (a). 10 m wind speed from CFSR and measurements calculated from a logarithm wind profile from Lidar measurements at 43 m, 50 m, 62 m, 82 m, and 100 m. (b). Wind direction from CFSR and Lidar measurement at 43 m. (c). Modeled significant wave height (solid lines) in comparison with Buoy measurement (black dotes), colored dotes show the absolute error. (d). Mean wave direction. (e). peak wave period. (f). mean wave period.



The time series of $H_{m0}$ at another 9 measurement sites, including Fjaltring (FG), Hanstholm (HM), A121 (A1), Vaderoarna (VA), Helgoland North (HN), Fino-1 (F1), Fino-3 (F3), Sleipner-A (SA), and Ekofisk (EK) in the North Sea during the two storm simulation are shown in Figure 7. The relative statistics are listed in Table 5. Considering the statistics of mean difference (BIAS), root mean square difference (RMSE), and scatter index (SI), WBLM generally gives better $H_{m0}$ than KOM and JANS

5   for most of the sites. However, WBLM tends to underestimate the largest waves during storm peaks, e.g. the storm peak at A121 (Figure 7c), Fino-1 (Figure 7f), and Ekofisk (Figure 7i).

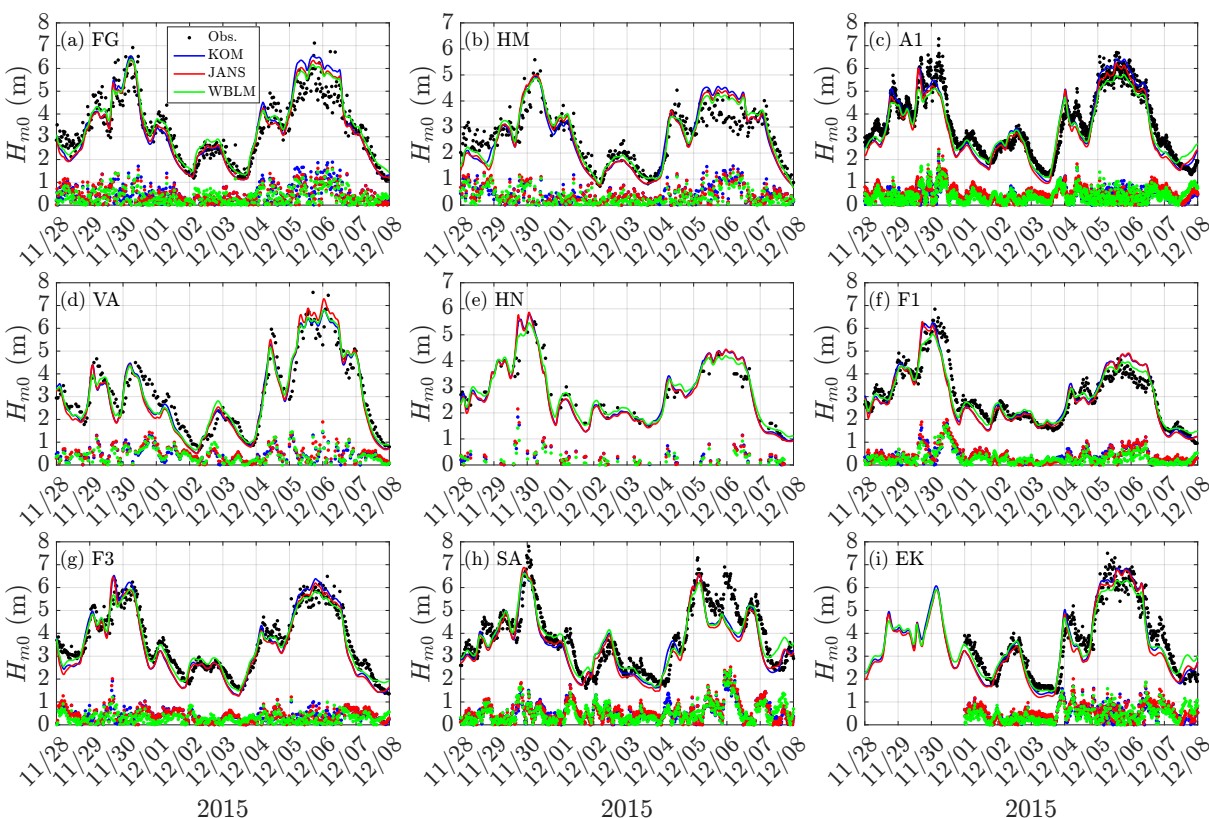

**Figure 7.** Time series of $H_{m0}$ at Fjaltring (FG), Hanstholm (HM), A121 (A1), Vaderoarna (VA), Helgoland North (HN), Fino-1 (F1), Fino-3 (F3), Sleipner-A (SA), and Ekofisk (EK) during the two winter storms in RUNE project. Observations are shown in black dots, modeled $H_{m0}$ using different source terms are shown in colored solid lines, and the corresponding colored dotes show the absolute error between modeled results and observations.

One-dimensional wave spectrum during the whole simulation period at RUNE site is presented in Figure 8. The colored lines in Figure 8a shows the data from buoy measurements, and the black envelop lines are used to mark the upper and lower bound of the measurement data. The colored lines in Figure 8b, 8c, and 8d are from SWAN simulations using KOM, JANS,

10   and WBLM source terms, and the black envelop lines are the same as in Figure 8a. The three simulations generally capture the shape of the measured spectrum. In comparison with the measurements, KOM and JANS tend to overestimate the energy



**Table 5.** Statistics of simulated significant wave height ($H_{m0}$) in comparison with measurements at Fjaltring (FG), Hanstholm (HM), A121 (A1), Vaderoarna (VA), Helgoland North (HN), Fino-1 (F1), Fino-3 (F3), Sleipner-A (SA), and Ekofisk (EK) from 2015-11-28 to 2015-12-08. The statistics include mean difference (BIAS), root mean square difference (RMSE), and scatter index (SI). In each column, the values of smallest absolute errors are signed with bold text.

| Statistics | Exp. | FG | HM | SA | EK | F1 | F3 | A1 | VA | HN |
|---|---|---|---|---|---|---|---|---|---|---|
| | KOM | **0.01** | -0.07 | **-0.12** | -0.12 | 0.06 | **-0.01** | -0.14 | -0.25 | -0.15 |
| BIAS | JANS | -0.06 | -0.08 | -0.27 | -0.12 | **0.02** | -0.08 | -0.22 | -0.32 | -0.26 |
| | WBLM | 0.13 | **0.00** | -0.19 | **-0.06** | 0.09 | **-0.01** | **-0.08** | **-0.11** | **-0.13** |
| | KOM | 0.64 | 0.54 | **0.51** | 0.54 | 0.49 | 0.51 | 0.47 | 0.69 | 0.60 |
| RMSE | JANS | 0.57 | 0.52 | 0.56 | 0.60 | 0.51 | 0.54 | 0.48 | 0.72 | 0.66 |
| | WBLM | **0.51** | **0.43** | 0.52 | **0.53** | **0.37** | **0.41** | **0.36** | **0.67** | **0.57** |
| | KOM | 0.19 | 0.20 | **0.14** | **0.17** | 0.17 | 0.17 | 0.13 | **0.17** | 0.16 |
| SI | JANS | 0.17 | 0.20 | **0.14** | 0.19 | 0.18 | 0.18 | 0.12 | **0.17** | 0.17 |
| | WBLM | **0.14** | **0.17** | **0.14** | **0.17** | **0.13** | **0.14** | **0.10** | **0.17** | **0.15** |

around the spectral peak while WBLM gives better energy estimation around the spectral peak. Both KOM and JANS show a level-off of energy at frequencies higher than about 0.3 Hz while the measurement and WBLM do not, which may explain the failure of KOM and JANS in simulating $T_{m01}$. However, seemingly WBLM tends to overestimate the energy at frequencies higher than the peak, which needs further investigations.

## 5  Discussion

This study first calibrates the WBLM wind-input and dissipation source terms in idealized cases, and further validates them in two real storm cases. In the selected cases, it is proven that the revised WBLM source terms can be used for real cases, and can provide certain wave properties better than the original ones in SWAN, such as KOM and JANS. However, long-term simulations and more comprehensive validations from different data resources such as satellite data are still necessary in further studies.

As mentioned in Du et al. (2017), one of the biggest strengths of WBLM is in the estimation of the air-sea momentum flux. Since this study mainly concerns its behavior in the wave simulations, the air-sea momentum flux (or roughness length / drag coefficient) is not included in the analysis. A future study with the focus on its momentum flux estimation was carried out and presented in Du (2017) Chapter 8, and it was found that the WBLM method provides reliable roughness length estimation in terms of the magnitude and the spatial distribution of it in coastal shallow water in comparison with point measurements and the Envisat ASAR backscatter.

The WBLM source terms is found to improve the prediction of the mean period significantly. Through analyzing the frequency spectra, it is speculated to be caused by an improved description of the high frequency part of the spectrum. However,



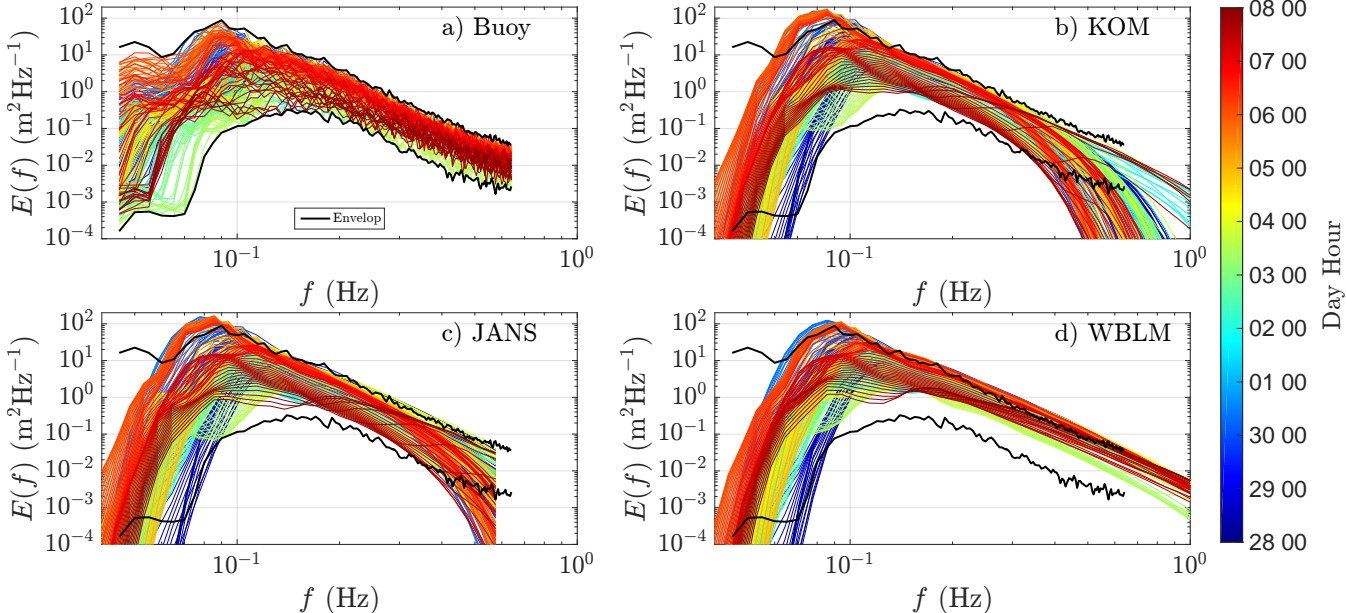

**Figure 8.** One-dimensional wave spectrum from buoy measurement at RUNE with all available data during the two storms (a), (b-d) are simulated with different source terms. The color of the lines represent different time. The black solid lines on each panel are the envelop lines to mark the upper and lower bound of the measurement data.

the energy from WBLM in high frequencies seems too high in comparison with measurements. Therefore, the energy distribution in high frequency range needs to be further investigated. One possible way of reducing this overestimated energy in high frequencies is by turning the parameters in the cumulative dissipation source function. However, the turning of these parameters have to be followed by the turning of the other parameters in the source terms including wind-input, white-capping, nonlinear four wave interactions, etc to make sure that both the stress estimation and the wave simulation are all satisfied.

Janssen (1991)'s wind-input source function was found by Du et al. (2017) and Du (2017) Chapter 5 to be wrongly implemented in SWAN. We tried our best to correct the code following the description of Bidlot et al. (2007) and by implementing some functions from WAM code (https://github.com/mywave/WAM) to SWAN. However, since this study is mainly focus on the usage of WBLM, a detailed analysis of Janssen implementation in SWAN is out of the scope of this paper.

## 6   Conclusions

This study aims at applying the WBLM source functions of Du et al. (2017) in SWAN for real wave simulations. Several improvements on the WBLM wind-input and white-capping dissipation source functions are realized. Firstly, the WBLM wind-input source function is modified by considering the wind profile change in the estimation of the non-dimensional critical height. Secondly, a revised white-capping dissipation source function is applied which enables the WBLM method being used





for varying wind conditions. Thirdly, a few refinements on the numerical algorithms of WBLM in SWAN are done to improve the model stability and efficiency, which make it possible to be used for large domain and high resolution simulations.

The new pair of WBLM wind-input and dissipation source function is calibrated with fetch-limited and depth-limited simulations. It is proven to be able to reproduce the benchmark wave growth curve of Kahma and Calkoen (1992), the energy level and the one-dimensional wave spectrum measured by Young and Babanin (2006b) in the depth-limited study.

The WBLM wind-input and dissipation source functions are validated with several point measurements during two storms over the North Sea. Results show that in comparison with the original wind-input and dissipation source functions in SWAN, namely Komen et al. (1984) and Janssen (1991), WBLM improves the prediction of significant wave height and mean wave period in comparison with measurements.

*Code and data availability.* The source code for SWAN used in this study is freely available at http:// swanmodel.sourceforge.net. The bathymetry data is obtained from EMODnet (http://www.emodnet-bathymetry.eu/). The observational wave data is downloaded from NOOS (https://noos.bsh.de/), FINO (http://fino.bsh.de), and eKlima (http://sharki.oslo.dnmi.no). CFSR 10 m wind speed is download from https://rda .ucar.edu/datasets/ds093.1/. The model data and source code modifications can be achieved from the corresponding author.

## Appendix A: D85 and JONSWAP spectra

The D85 (Donelan et al., 1985) spectra is described by:

$$E(f) = \alpha_D \frac{g^2}{(2\pi)^4} f_p^{-1} f^{-4} exp\left[-\left(\frac{f}{f_p}\right)^{-4}\right] \cdot \gamma_D^{exp\left[\frac{-(f-f_p)^2}{2\sigma_D^2 f_p^2}\right]}, \tag{A1}$$

where $f$ is the frequency and $f_p$ is the frequency at the spectral peak. $\alpha_D$ is a equilibrium range parameter which is written as:

$$\alpha_D = 0.006 \left(\frac{U_{10}}{c_p}\right)^{0.55}, \tag{A2}$$

where $U_{10}$ is 10 m wind speed. $c_p$ is the phase velocity at the spectral peak. In deep water condition, $c_p = \frac{g}{2\pi f_p}$, where g is the gravity acceleration. $\gamma_D$ is a peak enhancement factor:

$$\gamma_D = MAX\left[1.7 + 6.0 \log\left(\frac{U_{10}}{c_p}\right), 1.7\right], \tag{A3}$$

$\sigma_D$ is a peak width parameter, which is written as:

$$\sigma_D = 0.008 \left[1 + 4\left(\frac{U_{10}}{c_p}\right)^{-3}\right], \tag{A4}$$

The JONSWAP (Hasselmann et al., 1973) spectra is described by:

$$E(f) = \alpha_J \frac{g^2}{(2\pi)^4} f^{-5} exp\left[-\frac{5}{4}\left(\frac{f}{f_p}\right)^{-4}\right] \cdot \gamma_J^{exp\left[\frac{-(f-f_p)^2}{2\sigma_J^2 f_p^2}\right]} \tag{A5}$$





where the equilibrium range parameter it is written as:

$$\alpha_J = 0.076\widetilde{x}^{-0.22}, \tag{A6}$$

where $(\widetilde{x} = xg/U_{10}^2)$ is a non-dimensional fetch, and $x$ is the fetch. Parameterization for the peak enhancement factor $(\gamma_J)$ for JONSWAP spectra is not provided by Hasselmann et al. (1973). According to Hasselmann et al. (1973), $\gamma_J$) is scatted between 1.5 to 6.0, with an average value of 3.3. So we use the same equation as D85 (equation A3) with a limit of $1.5 \leq \gamma_J \leq 6.0$. The

5   peak width parameter is written as:

$$\sigma_J = \begin{cases} 0.07; f < f_p \\ 0.09; f \geq f_p. \end{cases} \tag{A7}$$

For both D85 and JONSWAP spectrum, the peak frequency $(f_p)$ for a given wind speed $(U_{10})$ and fetch $(x)$ is calculated from the fetch-limited wave growth relationship of Kahma and Calkoen (1992):

$$f_p = 2.1804\widetilde{x}^{-0.27} \cdot \frac{g}{U_{10}}. \tag{A8}$$

**Appendix B: Definition of statistics**

Take $X$ as observation value and $Y$ as modeled value, the mean difference is defined as:

$$BIAS = \frac{1}{N}\sum_{i=1}^{N}(Y - X)_i. \tag{B1}$$

10   The root mean square difference is defined as:

$$RMSE = \sqrt{\frac{1}{N}\sum_{i=1}^{N}(Y - X)_i^2}. \tag{B2}$$

The scatter index is defined as:

$$SI = \frac{\sqrt{\frac{1}{N}\sum_{i=1}^{N}(Y - X - BIAS)_i^2}}{\frac{1}{N}\sum_{i=1}^{N}|X_i|}. \tag{B3}$$

*Acknowledgements.* This project has received funding from the Danish Forskel project X-WiWa (PSO-12020) and the European Union's H2020 Programme for Research, Technological Development and Demonstration under Grant Agreement No: H2020-EO-2016-730030-CEASELESS. We are grateful to Jean Raymond Bidlot from ECMWF, Anna Rutgersson from Uppsala University, and Henrik Bredmose

15   from DTU Wind Energy for helpful discussions and inputs. Furthermore, we would like to thank Rogier Floors from DTU Wind Energy for providing the measurement data during Danish ForskEL project RUNE (12263).




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
