# Peer review of "Wave boundary layer model in SWAN revisited"

_Ocean Science, 2018_

## Referee Comment (RC1) · Anonymous Referee #1 · 5 Oct 2018

The paper "Wave boundary layer model in SWAN revisited" touches an interesting topic, which deserves further attention. However the difficulty of simultaneously improving the wind input and white capping dissipation source functions should be stressed right from the beginning since the result comes from the difference of these two terms and it is always a weak point in an analysis to deal with difference quantities. Moreover this difference, based on the parameterized physics, goes together with the numerical diffusivity associated to the discretized equations. Therefore my suggestion would be to discuss right from the start the difficulty in adjusting a difference of terms which is also conditioned by the numerical diffusivity.

Regarding the approach mention is made of fetch limited and depth limited studies. Since the adjustment is clearly related to wave age I think some more analysis about the duration limit conditions or the state of wave development would be in order to enhance the value of the paper.

[Figure]

When mentioning (page 2, paragraph 30) the "parameters" for tuning, mention should be made of which parameters. In here and in several other instances in the manuscript it would be important to link the parameters and the adjustment to their physical meaning so as to make the argumentation behind more solid and suitable for extrapolation to other cases.

In section 2.1, in paragraph 25, mention is made of how to generalize the phase velocity for the case of misalignment between wind and wave direction. This is an important point since the directional dispersion plays an important role. Some more analysis about the effect of directionality and how this would apply to for instance slant waves would be beneficial.

When presenting the equations for the wind input source function it is mentioned that the model tends to underestimate wave growth at lower frequencies and then the wave age tuning parameter form WAM is proposed. Some discussion showing how this element from WAM can be transferred directly to SWAN would also be in order.

When introducing the mean frequency and wave number following Bidlot, to put more emphasis on the high frequencies, this should be discussed together with the over/under dissipation at these high frequencies.

When replacing (page 5) fp in equation 17 by 0.86 <f> this should be discussed showing how it applies to a variety of cases where there may be more than one fp (e.g. by modal waves) or where fp is not a robust estimate.

In section 2.3 mention is made of a threshold so that the WBLM solves the energy within that interval in the wave spectrum. Some discussion about the sensitivity to that threshold and the implications that it has for very long or very short waves should be included. Particularly together with the selection of the numerical algorithm which also has implications for the numerical diffusivity.

In the case study for the North Sea some more discussion about the fetch, duration

and depth limitation for these two storms would be of interest.

When presenting in section 3.3 the NCEP Climate Forecast System version 2 for the wind forcing, some discussion of why these wind fields were selected and the uncertainty introduced by the wind selection should also be discussed, particularly related to the horizontal discretization of the atmospheric model and how that relates to the higher frequency components in the atmosphere and in the ocean.

When presenting the idealized fetch-limited study (section 4.1) a suggestion is made to use $U'/10$. Another possibility would be to scale with a value different from 10 meters/second, related to a length scale divided over a time scale. That would add a bit more of generality.

In this same section when increasing the number of parameters some discussion about the gain due to the higher number of fit parameters and the difficulty in application should be included.

Throughout the paper, particularly near the end, the advances obtained from the application of the WBLM should be strengthened, analysing the situation where the physics suggest there will be an improvement due to the explicit consideration of the wave boundary layer. This could serve to explain why the model underestimates high waves. This could also be related to the importance that wave-wave interactions play in such high waves, conditioned not only by the input and dissipation source terms. In this same line the underestimation of Tp could be explained from the physics, particularly stressing the wave boundary layer model dependence on frequency.

Finally in the discussion and conclusion section the wave boundary layer model advances should be discussed more in terms of the physics, linking the various descriptive paragraphs that appear now. That would strengthen the application and extrapolation of the proposed model.

A final small remark is that the paragraph (last paragraph, in section 5) explaining how

the Janssen wind-input source function was wrongly implemented in SWAN should be related to the rest of the paper, explaining how such an improved implementation leads to increase the robustness of the presented results if that is indeed the case.
* * *

---

## Editor Comment (EC1) · M. Espino Infantes (Editor) · 14 Oct 2018

Overview:

In this paper the authors extend the study presented in Du et al. (2017), to make the Wave Boundary Layer Model (WBLM) developed there for SWAN model, applicable for real wave simulations. In order to do that, several improvements on the WBLM wind-input and white-capping dissipation source functions are realized:

a) First, the WBLM wind-input source function is modified by considering the wind profile change in the estimation of the non-dimensional critical height

b) Second, a new white-capping dissipation source function is applied which enables WBLM methods being used for varying wind conditions

And finally, several improvements are made to the numerical WBLM algorithm , which

the authors explain that increase the model's numerical stability and computational efficiency.

This new WBLM is calibrated and validated again theoretical and real examples (in particular, during two North Sea storms) and it shown better performance in the simulations of Hs and Tz than the original source term.

General comments:

In my opinion, the first sections of the paper (Introduction, methods, experiments and results) are generally clearly written and readable. However, discussion and conclusion chapters should be expanded and improved. In particular, WBLM should be discussed more there in terms of the physics. Some ideas in order to discuss could be:

1. Page 3, equation 5: the possible effect of directional dispersion

2. Page 4, equation 11: introduction of the wave age tuning parameter

3. Page 4, equation 13: the possible effect of over/under dissipation at high frequencies

4. Page 5, lines 17-18: £How this applies to cases like modal waves?

5. Page 10, equation 22: the role of to scale U' with 10 m/s

Overall this paper needs moderate revision before acceptance for publication

Detailed comments:

1. Page 6, lines 17-18: Include some table comparing calculation times in the same examples for the WBLM (new and previous version), KOM and JANS formulations

2. Pag.10, line 25: Change Tp for fp

3. Figure 2: (title) Change Tp for fp

---

## Author Response (AR1)

**Point-by-point response to the reviews**

**Reply to Anonymous Referee #1**

The authors are sincerely grateful for the valuable comments and suggestions from the anonymous referee. These comments and suggestions brought some interesting discussions and helped improving the presentation of the paper. We give our response to these comments in the following, point-by-point.

**Comments:** The paper "Wave boundary layer model in SWAN revisited" touches an interesting topic, which deserves further attention. However the difficulty of simultaneously improving the wind input and white capping dissipation source functions should be stressed right from the beginning since the result comes from the difference of these two terms and it is always a weak point in an analysis to deal with difference quantities. Moreover this difference, based on the parameterized physics, goes together with the numerical diffusivity associated to the discretized equations. Therefore my suggestion would be to discuss right from the start the difficulty in adjusting a difference of terms which is also conditioned by the numerical diffusivity.

**Reply:** We added some discussion about the difficulty in adjusting the difference of source and sink terms as well as the numerical diffusivity in the last paragraph of Section 1[①].

**Comments:** Regarding the approach mention is made of fetch limited and depth limited studies. Since the adjustment is clearly related to wave age I think some more analysis about the duration limit conditions or the state of wave development would be in order to enhance the value of the paper.

**Reply:** Thanks for the suggestion. The duration limited study is relevant and needs further investigation. To the best of our knowledge, measurements about duration limited wave growth is rare and therefore it is difficult to evaluate the quality of the source terms. That is why we choose to use the fetch limited study instead of duration limited study. And the fetch limited study also reflects the state of wave development through fetch.

**Comments:** When mentioning (page 2, paragraph 30) the "parameters" for tuning, mention should be made of which parameters. In here and in several other instances in the manuscript it would be important to link the parameters and the adjustment to their physical meaning so as to make the argumentation behind more solid and suitable for extrapolation to other cases.

**Reply:** We specified the physics meaning of the two tuning parameters in the relative sentence in Section 1 (page 2, lines 29-30 and page 21, line 1)[②].

**Comments:** In section 2.1, in paragraph 25, mention is made of how to generalize the phase velocity for the case of misalignment between wind and wave direction. This is an important point since the directional dispersion plays an important role. Some more analysis about the effect of directionality and how this would apply to for instance slant waves would be beneficial.

**Reply:** Equation (5) $c=u(z_c)\cos(\theta-\theta_w)$ is not to generalize the phase velocity, but to define the "critical height" according to Miles (1957). Considering the misalignment between wind and wave direction, the "critical height" is the height where the phase velocity equals to the wind velocity component in the same direction as the phase velocity. The same method is also used by Janssen (1991), which is equation (3) in this paper. This equation applies to all the directions (36 directions in this study) in SWAN. We changed the expression in Section 2.1 (page 4, lines 3-5)[③] in case of misleading the readers.

**Comments:** When presenting the equations for the wind input source function it is mentioned that the model tends to underestimate wave growth at lower frequencies and then the wave age tuning parameter form WAM is proposed. Some discussion showing how this element from WAM can be transferred directly to SWAN would also be in order.

**Reply:** It is the same method (equation 3 in Bidlot, 2012) as that used in WAM. A positive wave age tuning parameter shifts the wave growth towards lower frequency. This paper is mainly about introducing the WBLM source terms for real applications, it is proved to be an effective way of tuning for both idealized cases and real storm cases, therefore we still keep using this parameter. We added some explanations in Section 2.1 (page 4, lines 13-14)[④].

**Comments:** When introducing the mean frequency and wave number following Bidlot, to put more emphasis on the high frequencies, this should be discussed together with the over/under dissipation at these high frequencies.

**Reply:** As discussed by Bidlot (2007), the introduction of the mean frequency and wave number with emphasis more on the high frequencies, is to reduce the impact of swell waves on the white-capping dissipation. Our present study is focus on the wind sea part as well, so it is reasonable to follow this method. More discussion is added in Section 2.2 (page 5, lines 1-2)[⑤].

**Comments:** When replacing (page 5) $f_p$ in equation 17 by 0.86 $<f>$ this should be discussed showing how it applies to a variety of cases where there may be more than one $f_p$ (e.g. by modal waves) or where $f_p$ is not a robust estimate.

**Reply:** Firstly, fp is a discretized variable in the wave model. That will make discontinuity when it is being use for parameterizing dissipation coefficient. Second, it will be difficult to determine which fp should be used in case of bimodal waves. The integrated variable <f> changes more gentle than fp and it always have one value in any given wave spectrum. Therefore, we prefer to use <f> instead of fp for generality and numerical stability. The uncertainty of using <f> in the bimodal wave case is not investigated in this study. Considering the model performs quite well in the two real storm simulations, we assume that the uncertainty is relatively small. We discussed this uncertainty in Section 2.2 (page 6, lines 3-6)[⑥].

**Comments:** In section 2.3 mention is made of a threshold so that the WBLM solves the energy within that interval in the wave spectrum. Some discussion about the sensitivity to that threshold and the implications that it has for very long or very short waves should be included. Particularly together with the selection of the numerical algorithm which also has implications for the numerical diffusivity.

**Reply:** In our previous description "energy containing frequency range" is vague which confuses the readers. We have accordingly changed it to "active frequency range" in Section 2.3 (page 6, lines 17-19)[⑦]. Although the maximum frequency is dynamically changing, only the frequencies whose contribution to the total wave stress is negligible are not calculated. So there is almost no influence to the result.

**Comments:** In the case study for the North Sea some more discussion about the fetch, duration and depth limitation for these two storms would be of interest.

**Reply:** More descriptions of the fetch, duration, and depth are added in Section 3.3 (page 10, lines 2-4 and lines 6-10)[⑧].

**Comments:** When presenting in section 3.3 the NCEP Climate Forecast System version 2 for the wind forcing, some discussion of why these wind fields were selected and the uncertainty introduced by the wind selection should also be discussed, particularly related to the horizontal discretization of the atmospheric model and how that relates to the higher frequency components in the atmosphere and in the ocean.

**Reply:** The CFSR wind shows good quality when evaluated with measurements and its quality has been proved to be good for wave simulations in the North Sea in many previous studies, e.g. Bolaños et al. (2014). The CFSv2 10 m wind as a horizontal resolution of bout 25 km and temporal resolution of 1 hour. In general wind conditions, over water, the difference in wind variability between a scale of 25 km to a few km is considered small. Therefore, the hourly CFSR data may be considered reasonable wind forcing. Though it may not be accurate in the presence of highly fluctuating wind on scales

smaller than 1 hour, e.g. Larsén et al. (2017). Relative explanations are add in Section 3.3 (page 8, lines 31-34 and page 9, line 1)[9].

**Comments:** When presenting the idealized fetch-limited study (section 4.1) a suggestion is made to use U'/10. Another possibility would be to scale with a value different from 10 meters/second, related to a length scale divided over a time scale. That would add a bit more of generality.

**Reply:** It is a good suggestion. We choose 10m/s because in this wind speed condition, the fetch-limited wave growth curves follows the reference quite well without needing to tun the dissipation coefficient. We appreciate your suggestion, and it will be subject of future testing, this stage we still keep using this parameter. We added some explanation in Section 4.1 (page 11, lines 11-13)[10].

**Comments:** In this same section when increasing the number of parameters some discussion about the gain due to the higher number of fit parameters and the difficulty in application should be included.

**Reply:** The number of parameters is discussed in Section 4.1 (page 11, lines 18-21 and page 12, lines 10-13)[11]. Use 4 parameters maybe easier to fit, but it requires more effort to use or change the parameters.

**Comments:** Throughout the paper, particularly near the end, the advances obtained from the application of the WBLM should be strengthened, analysing the situation where the physics suggest there will be an improvement due to the explicit consideration of the wave boundary layer. This could serve to explain why the model underestimates high waves. This could also be related to the importance that wave-wave interactions play in such high waves, conditioned not only by the input and dissipation source terms. In this same line the underestimation of Tp could be explained from the physics, particularly stressing the wave boundary layer model dependence on frequency.

**Reply:** All these points are very relevant and some discussions about the advances and limitations of the application of WBLM, and the role of nonlinear four wave interactions and swell dissipation in low frequency waves is added in the first and second paragraph of the Section 5[12].

**Comments:** Finally in the discussion and conclusion section the wave boundary layer model advances should be discussed more in terms of the physics, linking the various descriptive paragraphs that appear now. That would strengthen the application and extrapolation of the proposed model.

**Reply:** Some discussion about this is added in the second paragraph of the Section 5[12].

**Comments:** A final small remark is that the paragraph (last paragraph, in section 5) explaining how the Janssen wind-input source function was wrongly implemented in SWAN should be related to the rest of the paper, explaining how such an improved implementation leads to increase the robustness of the presented results if that is indeed the case.

**Reply:** Some explanation about the correction of the code is added in the last paragraph of Section 5[13]. We believe it is important to report this at might be useful to other SWAN users.

**Reply to M. Espino Infantes (Editor)**

We are grateful for the valuable comments and suggestions. We give our response to the editor's comments in the following.

**General comments**

**Comments:** Page 3, equation 5: the possible effect of directional dispersion

**Reply:** Equation (5) $c = u(z_c)\cos(\theta - \theta_w)$ is to define the "critical height" according to Miles (1957). Considering the misalignment between wind and wave direction, the "critical height" is the height where the phase velocity equals to the wind velocity component in the same direction as the phase velocity. The same method is also used by Janssen (1991), which is equation (3) in this paper. This equation applies to all the directions (36 directions in this study) in SWAN. We changed the expression in Section 2.1 (page 4, lines 3-5)[3] in case of misleading the readers.

**Comments:** Page 4, equation 11: introduction of the wave age tuning parameter

**Reply:** It is the same method (equation 3 in Bidlot, 2012) as that used in WAM. A positive wave age tuning parameter shifts the wave growth towards lower frequency. This paper is mainly about introducing the WBLM source terms for real applications, it is proved to be an effective way of tuning for both idealized cases and real storm cases, therefore we still keep using this parameter. We added some explanations in Section 2.1 (page 4, lines 13-14)[4].

**Comments:** Page 4, equation 13: the possible effect of over/under dissipation at high frequencies

**Reply:** As discussed by Bidlot (2007), the introduction of the mean frequency and wave number with emphasis more on the high frequencies, is to reduce the impact of swell

waves on the white-capping dissipation. Our present study is focus on the wind sea part as well, so it is reasonable to follow this method. More discussion is added in Section 2.2 (page 5, lines 1-2)[5].

**Comments:** Page 5, lines 17-18: £How this applies to cases like modal waves?

**Reply:** Firstly, fp is a discretized variable in the wave model. That will make discontinuity when it is being use for parameterizing dissipation coefficient. Second, it will be difficult to determine which fp should be used in case of bimodal waves. The integrated variable <f> changes more gentle than fp and it always have one value in any given wave spectrum. Therefore, we prefer to use <f> instead of fp for generality and numerical stability. The uncertainty of using <f> in the bimodal wave case is not investigated in this study. Considering the model performs quite well in the two real storm simulations, we assume that the uncertainty is relatively small. We discussed this uncertainty in Section 2.2 (page 6, lines 3-6)[6].

**Comments:** Page 10, equation 22: the role of to scale U' with 10 m/s

**Reply:** We choose 10m/s because in this wind speed condition, the fetch-limited wave growth curves follows the reference quite well without needing to tun the dissipation coefficient. We appreciate your suggestion, and it will be subject of future testing, this stage we still keep using this parameter. We added some explanation in Section 4.1 (page 11, lines 11-13)[10].

**Detailed Comments**

**Comments:** Page 6, lines 17-18: Include some table comparing calculation times in the same examples for the WBLM (new and previous version), KOM and JANS formulations

**Reply:** Suggestion taken. A table of calculation times during the idealized fetch-limited study using the new and old WBLM, KOM, and JANS source terms is added in Section 2.3. Relative description about the table is added in Section 2.3 (page 6, line 25-26 and page 7, line 1-3)[14].

**Comments:** Page.10, line 25: Change Tp for fp

**Reply:** Correction made. Now it is in Section 4.1 (page 11, line 29)[15].

**Comments:** Figure 2: (title) Change Tp for fp

**Reply:** Correction made in page 13 Figure 2[16].

**List of changes relevant to reviews**

①    The last paragraph of Section 1
②    Section 1 (page 2, lines 29-30 and page 21, line 1)
③    Section 2.1 (page 4, lines 3-5)
④    Section 2.1 (page 4, lines 13-14)
⑤    Section 2.2 (page 5, lines 1-2)
⑥    Section 2.2 (page 6, lines 3-6)
⑦    Section 2.3 (page 6, lines 17-19)
⑧    Section 3.3 (page 10, lines 2-4 and lines 6-10)
⑨    Section 3.3 (page 8, lines 31-34 and page 9, line 1)
⑩    Section 4.1 (page 11, lines 11-13)
⑪    Section 4.1 (page 11, lines 18-21 and page 12, lines 10-13)
⑫    The first and second paragraph of the Section 5
⑬    The last paragraph of Section 5
⑭    Table added in Section 2.3
⑮    Section 4.1 (page 11, line 29)
⑯    Page 13 Figure 2 caption

**Other minor changes**

(1) Page 1, line 22, "wave breaking" is changed to "wave-breaking"

(2) Page 2, line 15, a wrong citation of "Donelan (2001)" is removed.

(3) Page 2, line 29, "flexible such" is corrected to "such flexible".

(4) Page 7, line 18-19, we changed the expression of Step 3 to make it more clear.

(5) Page 11, line 10, "surmise" is changed to "speculate".

(6) Page 11, line 33, "$U_{10}$=05" is changed to "$U_{10}$=5".

(7) Page 12, line 17, "Our Model" is changed to "The results form WBLM source term" to be more specific.

(8) Page 18, line 5, added "in contrast to RUNE" and "in the open ocean sites", in Page 18, line 6, "Fino-1 (Figure7f)" is replaced with "Sleipner-A(Figure 7g)" to put more emphasis on the difference between coastal area and open ocean.

(9) Page 19, line 6, added "fetch-limited" to be more specific.

(10) Page 19, line 7, replace "them in" with "the model in idealized depth-limited cases and" to be more specific.

(11) Page 21, line 1-2, "turning" is corrected to "tuning".

(12) Page 21, line 1, added "(e.g. $C_{cu}$ in euqation 19)" to be more specific.

(13) Page 23, acknowledgments, one more funding is acknowledged.

[revised manuscript text omitted]